# Interstitial Control-Released Polymer Carrying a Targeting Small-Molecule Drug Reduces PD-L1 and MGMT Expression in Recurrent High-Grade Gliomas with TMZ Resistance

**DOI:** 10.3390/cancers14041051

**Published:** 2022-02-18

**Authors:** Ching-Ann Liu, Wei-Hsiu Liu, Hsin-I Ma, Yuan-Hao Chen, Dueng-Yuan Hueng, Wen-Chiuan Tsai, Shinn-Zong Lin, Horng-Jyh Harn, Tzyy-Wen Chiou, Jen-Wei Liu, Jui-Hao Lee, Tsung-Lang Chiu

**Affiliations:** 1Bioinnovation Center, Buddhist Tzu Chi Medical Foundation, Hualien 97002, Taiwan; anita8412@tzuchi.com.tw (C.-A.L.); shinn-zong@tzuchi.com.tw (S.-Z.L.); duke1945@tzuchi.com.tw (H.-J.H.); 2Department of Medical Research, Hualien Tzu Chi Hospital, Hualien 97002, Taiwan; 3Department of Neurosurgery, Neuro-Medical Scientific Center, Hualien Tzu Chi Hospital, Buddhist Tzu Chi Medical Foundation, Hualien 97002, Taiwan; 4Department of Neurological Surgery, Tri-Service General Hospital, National Defense Medical Center, Taipei 11490, Taiwan; doc20444@mail.ndmctsgh.edu.tw (W.-H.L.); tsghns01@mail.ndmctsgh.edu.tw (H.-I.M.); chenyh178@gmail.com (Y.-H.C.); hondy2195@yahoo.com.tw (D.-Y.H.); 5Department of Surgery, School of Medicine, National Defense Medical Center, Taipei 11490, Taiwan; 6Graduate Institute of Medical Sciences, National Defense Medical Center, Taipei 11490, Taiwan; 7Department of Biochemistry, National Defense Medical Center, Taipei 11490, Taiwan; 8Department of Pathology; Tri-Service General Hospital, National Defense Medical Center, Taipei 11490, Taiwan; doc31779@mail.ndmctsgh.edu.tw; 9School of Medicine, Tzu Chi University, Hualien 97002, Taiwan; 10Department of Pathology, Hualien Tzu Chi Hospital, Buddhist Tzu Chi Medical Foundation, Tzu Chi University, Hualien 97002, Taiwan; 11Department of Life Science and Graduate Institute of Biotechnology, National Dong Hwa University, Hualien 974, Taiwan; twchiou@gms.ndhu.edu.tw; 12Everfront Biotech Inc., New Taipei City 221, Taiwan; coldwee@efbiotech.com (J.-W.L.); juihaolee@efbiotech.com (J.-H.L.)

**Keywords:** recurrent glioblastoma, target therapy, clinical trial, intraparenchymal implantation, PD-L1, MGMT, temozolomide resistance, Cerebraca wafer, (Z)-*n*-butylidenephthalide

## Abstract

**Simple Summary:**

This study reports a potential new drug—Cerebraca wafer—that is designed to deliver its active pharmaceutical ingredient, (Z)-*n*-butylidenephthalide (BP), directly into the surgical cavity created when a brain tumor is resected. The therapeutic mechanism of Cerebraca wafer was shown to involve the following: (1) an IC_50_ of BP against tumor stem cells four times lower than that of bis-chloroethylnitrosourea (BCNU); (2) a synergistic effect between BP and temozolomide (TMZ), as demonstrated by a reduction in O^6^-methylguanine-DNA-methyltransferase (MGMT) expression level; (3) BP inhibition of programmed cell death-ligand 1 (PD-L1) protein levels, thereby activating T-cell cytotoxicity and increasing interferon-gamma (IFN-γ) secretion. The implantation of Cerebraca wafer is safe, no drug-related adverse events (AEs) and serious AEs (SAEs) were found. The median overall survival (OS) of patients receiving high-dose Cerebraca wafer have exceeded 17.4 months, and a 100% progression-free survival (PFS) rate at six month was achieved. In sum, these findings demonstrate that the Cerebraca wafer has superior therapeutic effects to Gliadel wafer in recurrent high-grade gliomas.

**Abstract:**

In recurrent glioblastoma, Gliadel wafer implantation after surgery has been shown to result in incomplete chemical removal of residual tumor and development of brain edema. Furthermore, temozolomide (TMZ) resistance caused by O^6^-methylguanine-DNA-methyltransferase (MGMT) activation and programmed cell death-ligand 1 (PD-L1) expression leads to immune-cold lesions that result in poorer prognosis. Cerebraca wafer, a biodegradable polymer containing (Z)-*n*-butylidenephthalide (BP), is designed to eliminate residual tumor after glioma resection. An open-label, one-arm study with four dose cohorts, involving a traditional 3 + 3 dose escalation clinical trial, of the Cerebraca wafer combined with TMZ on patients with recurrent high-grade glioma, was conducted. Of the 12 patients who receive implantation of Cerebraca wafer, there were no drug-related adverse events (AEs) or serious AEs (SAEs). The median overall survival (OS) of patients receiving low-dose Cerebraca wafer was 12 months in the group with >25% wafer coverage of the resected tumor, which is longer than OS duration in previously published studies (Gliadel wafer, 6.4 months). Patients who received high-dose Cerebraca wafer treatment had not yet died at the data cut-off date; a 100% progression-free survival (PFS) rate at six month was achieved, indicating the median OS of cohort IV was more than 17.4 months. In vitro study of the primary cells collected from the patients revealed that the IC_50_ of BP against tumor stem cells was four times lower than that of bis-chloroethylnitrosourea (BCNU). A synergistic effect between BP and TMZ was demonstrated by a reduction in MGMT expression. Furthermore, BP inhibited PD-L1 expression, thereby activating T-cell cytotoxicity and increasing interferon-gamma (IFN-γ) secretion. The better therapeutic effect of Cerebraca wafer on recurrent high-grade glioma could occur through re-sensitization of TMZ and reduction of PD-L1.

## 1. Introduction

Glioblastoma multiforme (GBM) involves multiple genetic mutations that result in high immunosuppression and resistance to chemotherapy and radiotherapy [1]. Introduced in 2005, the Stupp protocol [2] comprises post-surgery chemo-radiotherapy and adjuvant temozolomide (TMZ) chemotherapy for glioblastoma and has shown effective treatment and prolonged patient survival. However, more than 90% of glioblastomas recur within 24 months after Stupp standard procedures, and approximately 50% of glioblastomas classified as O^6^-methylguanine-DNA methyltransferase (MGMT) unmethylation, which are resistant to TMZ therapy [3,4], result in poor prognosis of patients with recurrent GBM. The MGMT promoter methylation status has been reported to be associated with patient survival. Around 95% of GBM patients that survived longer than 30 months after treatment are MGMT methylated; while only 36% of control patients (surviving for less than 30 months) are MGMT methylated [5].

In patients with recurrent GBM, the percentage of programmed death ligand 1 (PD-L1; diffuse/fibrillary PD-L1 expression: PD-L1 detected in non-necrotic areas) expression was 72.2%, whereas the percentage of patients with moderate or high expression of PD-L1 (5% or more) was 16.7% [6]. In recent years, immune checkpoint inhibitors targeting PD-L1, such as pembrolizumab and nivolumab, have significantly improved outcomes in melanomas, non-small cell lung cancers, and Hodgkin lymphomas, but have not shown favorable effects in postoperative adjuvant therapy for glioblastomas [7]. The correlation between poor response to immunotherapies and immunosuppressive tumor microenvironments (TME) has been reported in glioblastomas [8]. Furthermore, converting the “cold” TME of GBM to a “hotter” TME could increase the effectiveness of immunotherapies in GBM [8]. Investigations which address this concept could create better treatments for GBM. The correlations between PD-1/PD-L1 and prognosis in GBM are still uncertain. According to the Cancer Genome Atlas (TCGA) and Chinese Glioma Genome Atlas (CGGA) database, a high expression of both PD-L1 and PD-1 negatively correlates with prognosis of patients (*p* = 0.0031 and *p* = 0.0253, respectively) [9,10]; however, in other studies [11,12], no significant correlations were reported.

The Cerebraca wafer is designed to deliver its active pharmaceutical ingredient (API), (Z)-*n*-butylidenephthalide (BP), directly into the surgical cavity created when a brain tumor is resected. It was inspired by the Gliadel^®^ wafer, the only Food and Drug Administration (FDA)-approved chemotherapeutic implant for use during neurosurgical resection. The Gliadel wafer is a BCNU-loaded carboxyphenoxypropane-sebacic acid (CPPSA) copolymer that has been used for interstitial chemotherapy in high-grade gliomas since the 1990s. In clinical practice, patients implanted with the Gliadel wafer exhibited a higher risk of complications with seizures and a significant increase in intracranial hypertension [13].

The excipient of the Cerebraca wafer is a biodegradable polyanhydride CPPSA copolymer (similar to that in the Gliadel wafer) that is safe and has been intracranially delivered to animals and patients for more than 20 years. Polyanhydrides are biodegradable polymers which have been developed since 1980 [14]. Toxicology studies have been performed on the rat and monkey brain through the implantation route [15,16]. In the rat study, 42 hemispheres of adult Sprague–Dawley rats’ brains received bilateral frontal lobe implantation of CPPSA. None of the animals showed any behavioral changes or neurological deficits [16]. In the monkey study, a group that received CPPSA brain implantation revealed no neurological or general deleterious effects [15]. The Gliadel wafer has been approved by the FDA since 1996, following the pivotal phase III clinical trial which included 120 GBM patients receiving Gliadel wafer and 120 patients receiving blank wafer (CPPSA only) as a placebo control [17]. These studies provided the safety information for the excipient of Cerebraca wafer.

Cerebraca wafer is a product of Everfront Biotech Inc. BP was chosen as the API due to its anticancer effects and its ability to reduce the expression of MGMT in glioma cells [18,19]. Each Cerebraca wafer is composed of 75 mg BP and 225 mg biodegradable excipient. The manufacturing procedures of Cerebraca wafer comply with good manufacturing practice (GMP) as defined by the U.S. FDA and the TFDA (Ministry of Health and Welfare, Taiwan). The in vitro release profile of Cerebraca wafer indicates that the drug can be slow-released for at least 21 days [19]. On exposure to the aqueous environment of the resection cavity, the anhydride bonds in the copolymer are hydrolyzed, releasing BP. Unlike Gliadel wafer, that contains only 3.8% of API, Cerebraca wafer contains 25% API, which provides a higher local concentration after implantation. This high local concentration enables the drug to achieve a greater diffusion distance. With respect to the poor diffusion (2 mm) that results in the insufficient clinical outcomes of the Gliadel wafer, Cerebraca wafer presents advantages of a high safety margin and long diffusion distance (20–50 mm). Under the approval of the U.S. FDA and TFDA (the Ministry of Health and Welfare, Taipei, Taiwan), phase I/IIa clinical trials of the Cerebraca wafer have been conducted in Taiwan. The present study reports the safety and preliminary results of a phase I clinical trial of the Cerebraca wafer intraparenchymal implantation in human recurrent high-grade glioma.

## 2. Materials and Methods

### 2.1. Study Design

This single-arm, open-level phase I clinical trial, designed as a 3 + 3 dose escalation, was approved by the TFDA and the Tzu Chi Research Ethics Committee and conducted in the Tzu Chi and Tri-service General Hospitals from November 2017 to June 2019. A total of 17 patients diagnosed with recurrent GBM were identified. After inclusion and exclusion criteria were applied, a final total of 12 patients were included. Three patients in the first cohort received one Cerebraca wafer implantation, three in the second cohort received two Cerebra wafer implantations, three in the third cohort received four Cerebraca wafers implantations, and three in the fourth cohort received six Cerebraca wafer implantations. TMZ was administered at 75 mg/m^2^/day for 42 days and a further 200 mg/m^2^/day for 5 days every 4 weeks thereafter. Functional evaluations included the Mini-Mental State Examination, Karnofsky Performance Score, and the European Organization for Research and Treatment of Cancer Quality of Life Questionnaire.

ClinicalTrials.gov [Internet]. Bethesda (MD): National Library of Medicine (US). 2000 Feb 29—Identifier NCT03234595, A Phase I/IIa Study of Cerebraca wafer Plus Adjuvant Temozolomide (TMZ) in Patients with Recurrent High Grade Glioma; 2017 Jul 31. Available from: https://clinicaltrials.gov/ct2/show/NCT03234595 (accessed on 15 February 2022)

The major inclusion and exclusion criteria are listed as follows:

Inclusion criteria

Female or male, age ≥20 years old.Patient has diagnosed recurrent high grade glioma, including anaplastic astrocytoma and glioblastoma multiforme.Patient has unilateral tumor in cerebrum that can be excised in one operation.Patient has recurrence of glioma.Patient has undergone standard therapy for their prior glioma episode; for patients with anaplastic astrocytoma, the prior standard therapy should include surgical resection, radiation and adjuvant temozolomide (or PCV [procarzine, lomustine and vincristine]); for patients with glioblastoma multiforme, the prior standard therapy should include surgical resection, radiation and adjuvant temozolomide.Patient has a Karnofsky Performance Score (KPS) ≥50.Patient is recovered from toxicities from prior systemic therapies and has adequate hematopoietic function at screening and before using study medication.Patient with no or mild organ impairment.Patient agrees not to use food or dietary supplements that contain *Angelica sinensis* from Screening Visit to Day 21.All male patients and female patients with child-bearing potential (between puberty and two years after menopause) should use appropriate contraception method(s) for at least four weeks after Cerebraca wafer treatment and TMZ treatment (whichever is longer).

Exclusion criteria

Patient has participated in other investigational studies within four weeks prior to receiving Cerebraca wafer.Patient with known or suspected hypersensitivity to Cerebraca wafer, TMZ or the excipient.Patient has tumor that cannot be surgically removed without significantly affecting vital function.Patient has external-beam radiation therapy within four weeks before study entry.Patient has immuno-compromised condition, or has a known autoimmune condition, or is human immunodeficiency virus (HIV) seropositive.Patient has on-going moderate to severe organ impairment other than study indication that may confound the efficacy evaluation, safety evaluation or usage of TMZ.

### 2.2. Imaging

Preoperative non-contrasted and contrasted MRI were performed and combined with functional MRI and/or fiber tractography (if lesions were involved or were adjacent to the motor cortex or corticospinal tract) on the morning of the operative day for navigational guidance of tumor excision. Postoperative MRI evaluation was performed on the second day, 1 month, 2 months, 3 months, and 6 months after operation. The longest diameter and the longest perpendicular diameter obtained from the MRI image are multiplied to estimate the tumor area according to the WHO guideline [20]. A similar method (longest diameter and the longest perpendicular diameter multiplied) was used to calculate the surface area of a wafer. The wafer coverage was determined by the wafer surface area/the tumor area.

### 2.3. Safety Evaluation

Hematological testing, biochemical testing, urinalysis, electrocardiography, and evaluation of BP and CPPSA serum concentrations were performed daily for 7 days during hospitalization and at every monthly follow-up for 6 months. The dose-limiting toxicity (DLT) determined by NCI-CTCAE 4.03, and physical and neurological examinations were conducted daily during hospitalization and monthly during every OPD visit over 6 months. Any AE or SAE was recorded and reported to the data and safety monitoring board.

### 2.4. Patient-Derived Primary Tumor Cultures

Proteolytic enzymes are widely used in cell dissociation. Papain has been proved less damaging and more effective than other proteases with some tissues. Papain has been used with fetal as well as postnatal brain tissue specimens to generate maximal dissociation and viability of neurons. Adult brain tumor samples were surgically obtained from patients with relapsed GBM who consented to tissue use under protocols approved by the Tzu Chi General Hospital Institutional Review Board (IRB108-15-A). The samples were dissected and dissociated to single-cell suspensions using the Worthington Papain Dissociation System (Cat. LK003150, Worthington Biochemical Corporation, Freehold, NJ, USA), including the reagents of EBSS, papain enzyme, DNase and albumin ovomucoid protease inhibitor. The tissues were removed and dissected following the manufacturer’s instructions. The tissue slices were collected in Ca/Mg-free Hanks’ balanced salt solution (Cat. 88284, Thermo Fisher Scientific, Waltham, MA, USA) and then cut into smaller pieces in Earle’s balanced salt solution (EBSS; 117 mM NaCl, 5 mM KCl, 1.8 mM CaCl_2_, 0.8 mM MgSO_4_, 26 mM NaHCO_3_, NaH_2_PO_4_-H_2_O, 5.56 mM glucose, and 0.03 mM phenol red). The tissue slices were then digested with papain (20 units/mL papain and 0.005% DNase/mL in 1 mM L-cysteine/0.5 mM ethylenediaminetetraacetic acid) with gentle shaking for 1.5 h at 37 °C. Following incubation, the dissociated cells were passed through a 40 mm cell strainer (Becton Dickinson, Mountain View, CA, USA) and spun at 300 g for 5 min in an Allegra-XR centrifuge (Beckman, Brea, CA, USA). The pellets were resuspended in EBSS with DNase I (100 units/mL) and albumin ovomucoid protease inhibitor (1 mg/mL) and carefully layered in a cell suspension on top of 5 mL of albumin ovomucoid protease inhibitor (1 mg/mL) to perform the discontinuous density gradient. The samples were spun at 70× *g* for 6 min, and the pellets were resuspended at 4 × 10^6^ cells/mL in Dulbecco’s Modified Eagle Medium/Nutrient Mixture F-12 containing 10% fetal bovine serum. The cells were plated at 1 mL/dish into 100 mm TC dishes (CLS430167, Corning, Grand Island, NY, USA) and incubated at 37 °C in 5% CO_2_.

### 2.5. RNA Extraction and Reverse Transcription from Total RNA

Cells were treated for 24 h with 400 µM BP followed by total RNA extraction from harvested cells with the RNeasy plus kit (Cat. 74134, Qiagen, Hilden, Germany), according to the manufacturer’s instructions. Total RNA was then reversely transcribed into cDNA using the SuperScript reverse transcriptase (Cat. 18080093, Thermo Fisher Scientific, Waltham, MA, USA), according to the manufacturer’s instructions. An amount of 2 mg total RNA was incubated with RNase-free DNase I (1U/µL) solution of DNase I (Cat. 79254, Qiagen, Hilden, Germany) for 1 h at 37 °C, DNase I inactivation with 2 uL of 25 mM ethylenediaminetetraacetic acid (EDTA) for 10 min at 65 °C, followed by incubation with 3 µL of random primers (N8080127, Thermo Fisher Scientific, Waltham, MA, USA) and 1 µL of 10 mM dNTPs (Cat. 201912, Qiagen, Hilden, Germany) for 5 min at 56 °C, ending with 4 °C, incubation with RNaseOUT (Cat. 10777019, Thermo Fisher Scientific, Waltham, MA, USA) for 2 min at room temperature. After additional incubation with 3 µL of reverse transcriptase (200 U) for 10 min at room temperature, the mixture was further incubated for 1 h at 42 °C and then for 15 min at 70 °C. Control of the reaction was carried out in parallel without adding reverse transcriptase. The resulting cDNA (50 ng/µL) was used to perform real-time PCR.

### 2.6. Real-Time Polymerase Chain Reaction (Real-Time PCR) Using SYBR Green

To quantify the MGMT expression, real-time PCR reaction was performed using Power SYBR Green PCR Master Mix (Cat. 4368702, Thermo Fisher Scientific, Waltham, MA, USA). The following primer sequences were used: MGMT forward: 5′-GCTGAATGCCTATTTCCACCA-3′, MGMT reverse: 5′-CACAACCTTCAGCAGCTTCCA-3′; 18S forward: 5′-CGGCTACCACATCC AAGGAA-3′, 18S reverse: 5′-GCTGGAATTACCGCGGCT-3′. Real-time PCR was performed at 95 °C for 10 min, then 45 cycles of 95 °C for 15 s, 60 °C for 20 s and 72 °C for 20 s. Gene expression was determined by normalization against 18S ribosomal RNA expression using the ΔΔCq method. In each separate experiment, each sample was analyzed in triplicate to confirm repeatability.

### 2.7. Flow Cytometry

The cell cycle distribution was analyzed using flow cytometry (CytoFlex flow cytometer, Beckman Coulter, Pasadena, CA, USA). The cells were seeded onto 10 cm dishes at 5 × 10^5^ cells per dish and incubated at 37 °C in 5% CO_2_ for 24 h, followed by an additional 24 h treatment of 400 µM BP. Cells were trypsinized, washed with PBS, and stained with PE-conjugated PD-L1 antibody (Cat. 393608, Biolegend, San Diego, CA, USA) for 30 min at RT before flow cytometry analysis. Control experiments were performed unstained, stained with PE-IgG, with or without compensation beads (Cat. 01-2222-42, Thermo Fisher Scientific, Waltham, MA, USA), or stained with PE-PDL-1, with or without compensation beads. A positive result was indicated by PD-L1 antibody with compensation beads control.

### 2.8. IFN-γ ELISA Assay

IFN-γ analysis was performed using the standard protocol outlined for the BioLegend ELISA kit (Cat. 430104, BioLegend, San Diego, CA, USA). The ELISA plate was coated with 100 µL anti-IFN-γ capture antibody and incubated overnight at 4 °C. The plates were washed four times using 300 µL/well of wash buffer (PBS with 0.05% Tween 20). To block non-specific binding and reduce background noise, 200 µL 1× Assay Diluent A was added per well. The plate was sealed and incubated at RT for 1 h on a plate shaker (500 rpm with a 0.3 cm circular orbit). The plate was then washed four times with wash buffer, after which 100 µL/well of standards or samples were added to the appropriate wells. The plate was sealed again and incubated at RT for 2 h on a plate shaker, followed by washing four times with wash buffer. Diluted detection antibody solution (100 µL) was then added to each well. The plate was once again sealed and incubated at RT for 1 h on a plate shaker. Afterwards, the plate was washed four times with wash buffer, and 100 µL of diluted Avidin-HRP solution was added subsequently to each well. The plate was sealed and incubated at RT for 30 min with shaking. The plate was washed five times with wash buffer, and 100 µL of freshly mixed TMB substrate solution was added. The plate was then incubated in the dark for 20 min until the reaction was stopped by adding 100 µL stop solution to each well. Absorbance was read at 450 and 570 nm within 15 min. Finally, the absorbance at 570 nm was subtracted from the absorbance at 450 nm.

### 2.9. MGMT Promoter Methylation Determination

MGMT promoter methylation status was evaluated by pyrosequencing [21]. In brief, the genomic DNA from patient tumor tissue was extracted, and then PCR was used to amplify the MGMT promoter region. The ssDNA isolated from the PCR product was used as a template for pyrosequencing. The pyrosequencing analysis was performed by QIAGEN PyroMark Q24.

## 3. Results

### 3.1. Patient Characteristics

Twelve patients (mean age 49.9 years, range 27.0–68.9 years) received brain tumor resection surgery and Cerebraca wafer implantation with TMZ administration (Table 1), which included five (41.7%) mesenchymal type and seven (58.3%) non-mesenchymal type gliomas. Ten patients (83.3%) had MGMT promoter unmethylation, which has been associated with anti-alkylating agent activity and poor survival after long-term chemotherapy.

### 3.2. Treatment Program

Cerebraca wafer implantation was performed after brain tumor resection. The patient flow diagram is shown in Figure 1, with patient accrual at two institutions shown as the number of patients by Cerebraca wafer dosing level.

### 3.3. Cerebraca Wafer Implantation Demonstrated a Good Safety Profile

The primary analysis was to determine the tolerability of the Cerebraca wafer (Figure 1). To determine the maximum tolerated Cerebraca wafer dose, a starting dose of 75 mg was administered to the first cohort, which was then increased to 150, 300, and 450 mg. During the dose escalation study, no dose-limiting toxicities (DLTs) were reported in any cohort. On the basis of this finding, the protocol-defined maximum tolerated dose (MTD) was reached (450 mg). No drug-related adverse events (AEs) or serious AEs (SAEs) were observed in this study. As Table 1 shows, AE ≥ Grade 3 included liver function abnormal, wound complications, sepsis, and lung infection, whereas neurologic AE ≥ Grade 3 included CSF leakage and hydrocephalus.

### 3.4. Cerebraca Wafer Implantation plus TMZ Administration Resulted in Residual Tumor Shrinkage after Surgical Resection

As Figure 2 shows, a patient treated locally with one Cerebraca wafer (right deep parietal) showed progressive recurrent tumor reduction over 6 months. On the opposite site, a small nodule without Cerebraca wafer implantation grew progressively, even under TMZ treatment.

### 3.5. Molecular Analysis of Primary Glioma Cell Lines Obtained from Surgery Revealed the Beneficial Effects of Cerebraca Wafer

Primary GBM cultures were obtained from the patients recruited for the clinical trial (8 out of 12). Subsequently, the following parameters were evaluated: TMZ resistance (as evaluated by MGMT pyrosequencing); cancer stem cell marker (CD133 and SOX2); half-maximal inhibitory concentrations (IC_50_) of BP, BCNU, and MTIC (the active form of TMZ); PD-L1 expression levels; and immune responses (co-culture with immune cells).

### 3.6. Cerebraca Wafer Reduced the Cancer Stem Cell Percentage in Recurrent GBM

As Table 2 shows, seven out of eight primary glioma lines acquired in this clinical trial revealed high expression of the cancer stem cell markers, CD133 and SOX2, suggesting that tumor recurrence presents high cancer stem cell generation. The IC_50_ of BCNU in T1, a non-tumor stem cell line, was approximately 1200 μM. We further showed that GBM cells with high stem cell marker expression had higher IC_50_ when treated with BCNU (≥1600 μM). This result possibly correlates to the limited therapeutic effects of the BCNU-loaded Gliadel wafer. However, the IC_50_ of BP in patient-derived primary glioma cell lines was lower than that of BCNU, indicating that the API of the Cerebraca wafer was more efficient than that of the Gliadel wafer. We further demonstrated that the IC_50_ of BP was around four times lower than that of the BCNU. Additionally, the difference in drug loading between the Cerebraca and Gliadel wafers was almost 10-fold (75 vs. 7.7 mg per wafer), which makes the cancer cell killing possible.

### 3.7. Cerebraca Wafer Provides a Synergistic Effect with the Alkylating Agents and Reduces Their Drug Resistance

Our results show that the IC_50_ of BP combined with MTIC was lower, showing that BP had an in-combination effect with TMZ. MGMT expression levels in patient-derived primary tumor cell lines were also found to be downregulated 22–93% by BP (Figure 3A). Interestingly, this downregulation of MGMT was further enhanced (80–98%) by the addition of MTIC, suggesting that Cerebraca wafer implantation, followed by TMZ administration, killed the residual tumor cells. We showed that after pretreatment, the IC_50_ of BP combined with MTIC decreased 50–66% compared with the non-pretreatment group. This result may reflect the real situation when Cerebraca wafer is implanted into the brain, and starts to release the API, BP. Twenty-four hours after Cerebraca wafer implantation, the administration of TMZ was initiated. The pretreatment with BP may have further enhanced the effect of TMZ.

PD-L1 expression levels of the patient-derived GBM cells were evaluated via flow cytometry. Moderate or high PD-L1 expression (5% or more) was detected in all eight primary glioma lines. Furthermore, after 24 h of BP treatment, PD-L1 expression decreased in glioma lines with more than 45% initial PD-L1 expression (Figure 3B). These results suggest that the implantation of Cerebraca wafer alters the glioma microenvironment. The interferon γ (IFN-γ) analysis also confirms this. In the co-cultures with immune cells, BP administration increased IFN-γ expression levels 1.63-fold, indicating a positive anti-tumor immune response (Figure 3C).

### 3.8. Preliminary Efficacy Analyses Suggest That Cerebraca Wafer Improved OS

Table 1 shows the OS and PFS rate in this study. The 100% 6M PFS was achieved in the high dose group, which was better than that for Avastin (44%) [22]. Patient survival analysis showed that cohort IV had better survival than cohorts I–III, suggesting that the dose level in cohort IV (six wafers, 450 mg) may be optimal in producing better clinical outcomes (Figure 3D). There was no drug-related AE or SAE in cohort I–IV, suggesting that a higher dose level may be provided to acquire a better clinical outcome. Further findings revealed that a wafer coverage of more than 25% of the resected tumor was an essential factor in the improvement of patient survival. As Figure 3E shows, patients who received a wafer coverage of more than 25% had better survival outcomes (12 months). Furthermore, it has been demonstrated that all three patients in cohort IV receiving six wafers had not died before submission of this manuscript, indicating the median OS of cohort IV was more than 17.4 months, as shown in Figure 3D.

### 3.9. Overall Survival Is Correlated with the Combination Effect of BP and TMZ

The relationship between data from the primary cultures and overall survival was analyzed. TMZ sensitivity was found to be downregulated by the regulation of MGMT expression. In most recurrent GBM cases, patients have developed chemoresistance to TMZ, thus resulting in reduced effectiveness of subsequent TMZ treatment. One patient died because of a lung embolism, while the other patients died from tumor progression. Excluding the patient who died from a lung embolism, the overall survival was correlated to the combination effect of BP plus MTIC, the active form of TMZ (Figure 3F). Although more evidence is required to increase the confidence of the findings, they indicate that overall survival is associated with sensitivity to BP + TMZ.

## 4. Discussion

Although BP had a lower IC_50_ than BCNU toward brain tumor primary cells (Table 2), it has been demonstrated that BP only targets cancer cells and shows low cytotoxicity toward normal cells, such as human bone marrow cells or mouse fibroblasts [23]. In addition, Cerebraca wafer has a higher therapeutic index than BCNU wafer. According to their MSDS (W333301, Sigma-Aldrich; C0400, Sigma-Aldrich, St. Louis, MO, USA), the oral LD_50_ of *n*-butylidenephthalide (which contains more than 85% BP) and the BCNU in rats are 1850 mg/kg and 20 mg/kg, which can be converted to 471 mg BP/rat and 6 mg BCNU/rat, respectively. We have previously demonstrated that 3 mg/rat of BP or BCNU containing wafer can suppress the RG2 brain tumor volume in more than 50% of experimental rats [19], which can be represented as ED_50_. The therapeutic indices, as indicated by LD_50_/ED_50_ of Cerebraca wafer and BCNU wafer, were approximately 157 and 2, respectively, suggesting that Cerebraca wafer is safer than BCNU wafer.

Local drug administration enables the delivery of higher drug concentrations and fewer systemic side effects. However, previous studies on Gliadel wafer implantation revealed the possibility of local complications, such as brain edema and hemorrhage [24,25,26]. In our phase I dose escalation study, no Cerebraca wafer-related brain edemas were found. Most of the SAEs reported in this clinical trial were TMZ-related or were due to disease progression. Furthermore, since brain edemas were rarely observed in patients who received Cerebraca wafer treatment, immune privilege can be avoided. At the same time, beneficial effects of Cerebraca wafer on the recurrent GBM patients were observed. In the high dose group, a 17.4-month median overall survival and a 100% 6-month PFS were achieved. This clinical outcome is better than the previous published result for Gliadel wafer, with 6.4-month median OS and Avastin, 9.3-month median OS, and 44% 6-month PFS observed.

Glioma stem cell populations have been found to increase after TMZ and radiation [27,28]. Of the eight primary cell lines collected, seven lines showed high expression of the stem cell markers, CD133 and SOX2. As demonstrated in our study, patients with higher stem cell marker expression have higher BCNU IC_50_. Furthermore, BP has previously demonstrated a lower IC_50_ than BCNU. The in vitro analysis of the patient-derived primary cell cultures revealed findings consistent with the IC_50_ of BP, which was four times lower than that of BCNU.

One Gliadel^®^ wafer contains 7.7 mg BCNU, and the maximum recommended dose is eight wafers, or a total of 61.6 mg BCNU [29]. Conversely, one Cerebraca wafer contains 75 mg of BP, and six wafers, or 450 mg, achieved MTD. This indicates that the therapeutic effect of BP is 4 times as great as for carmustine while the dose of the drug is 7.3 times. In conclusion, the therapeutic effect of the Cerebraca wafer would be nearly 29.2 times higher than that of the Gliadel wafer.

As Figure 3E shows, the overall survival in patients who had more than 25% wafer coverage of the residual tumor intersection showed the highest overall survival outcomes. Grades III and IV gliomas typically recur within 2 cm of the original location [30]. To provide enough API to kill residual tumors, the coverage percentage of Cerebraca wafer was set as a criterion for better clinical outcomes.

A meta-analysis report of nine clinical trials, involving 806 patients with GBM, reported that PD-L1 expression in tumor tissues was significantly related to a poor OS [31]. The downregulation of the JAK/STAT pathway and immune response in recurrent tumors showed that the GBM area becomes an immunologically cold tumor exhibiting immune protection [32]. Given the failure of immunotherapy in GBM due to the paucity of GBM-infiltrating T cells, the concept of turning “cold” tumors into “hot” tumors has been proposed—increased PD-L1 expression is one of the major reasons for the failure of immunotherapy. In our study, PD-L1 was reduced by BP in six out of eight patient primary glioma lines (Table 2). We have further shown that this reduced PD-L1 can increase IFN-γ 1.6-fold via the PBMC immune response to tumor cells. The higher PFS observed in that study showed that the therapeutic strategy of Cerebraca wafer combined with CIK may produce better clinical outcomes in GBM. Locally reducing PD-L1 expression with BP via Cerebraca wafer implantation might not only enhance tumor microenvironment immunity but also avoid the development of systemic cytokine storms.

Molecular mechanisms including downregulation of MGMT and PD-L1, and killing cancer stem cells utilizing BP, are reported in this study. It has previously been reported that *n*-butylidenephthalide exerted suppressive effects on tumor cell proliferation in human, rat, and mice GBM cell lines via downregulation of cell-cycle regulators and increase in apoptosis-associated genes and proteins [33,34,35]. Moreover, in vivo results indicated that the subcutaneous injection of *n*-butylidenephthalide in xenograft mouse models, not only suppressed human GBM tumor growth, but also prolonged survival rates [34,35]. *n*-butylidenephthalide induces anti-tumor activity through the following effects: (1) promotion of senescence in GBM cells associated with its function in telomerase, Skp2, and p16 regulation [34,35]; (2) induction of tumor cell apoptosis via the upregulation of Nur77, an orphan nuclear receptor [33]; (3) inhibition of the expression of Axl, an essential regulator in cancer metastasis, thereby reducing the migratory and invasive capabilities of glioma cells [36]; (4) eradication of tumor stem cells by downregulating oncogenes, such as SOX2 and OCT4 [37]; (5) reversal of TMZ resistance by suppressing MGMT mRNA and protein expression [36], and (6) downregulation of glioma PD-L1 expression levels to induce anti-tumor immune responses with DNMT3b activation [38]. Because the tested materials used in these studies were commercially available *n*-butylidenephthalide, in which more than 85% were the Z-form geometric isomer (BP), this implies that BP played a major role in the beneficial effects toward brain tumor treatment observed. In addition, the pleiotropic effects of BP might be due to the activation of the AMPK pathway [39], its downstream AXL receptor [36], and DNMT regulation [40].

In the present study, BP was demonstrated to be a novel small-molecule drug that can target high-grade gliomas through a maximum of six wafers (75 mg per wafer) without obvious SAEs. Unlike that of BCNU, the IC_50_ of BP was found to be the same to tumor stem cells as to non-stem tumor cells. BP can suppress PD-L1 expression in malignant glioma cells by locally implanted polymer, thereby avoiding a systemic cytokine crisis while chemically eradicating residual tumor cells. Furthermore, these effects are not limited to gliomas. Although we showed an exciting result of Cerebraca wafer treatment in this phase I study, further clinical trials that include more patients to achieve an effective sample size are required. All in all, these findings demonstrate that the Cerebraca wafer has superior therapeutic effects to the Gliadel wafer in recurrent high-grade gliomas.

## 5. Conclusions

We showed that Cerebraca wafer has superior therapeutic effects to the Gliadel wafer in recurrent high-grade gliomas. Since no drug-related AEs or SAEs were observed, a higher dose may be considered for further investigation. The better therapeutic effect of Cerebraca wafer may occur through resensitization of TMZ and reduction of PD-L1.

## Figures and Tables

**Figure 1 cancers-14-01051-f001:**
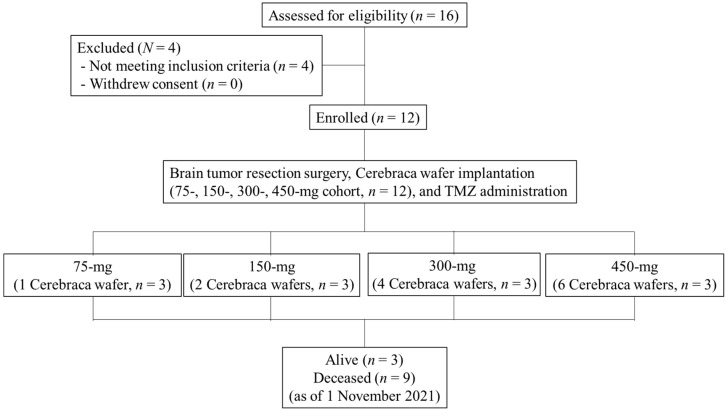
Study CONSORT diagram. Flowchart shows the number of patients who were screened, enrolled into the treatment groups, and completed the study.

**Figure 2 cancers-14-01051-f002:**
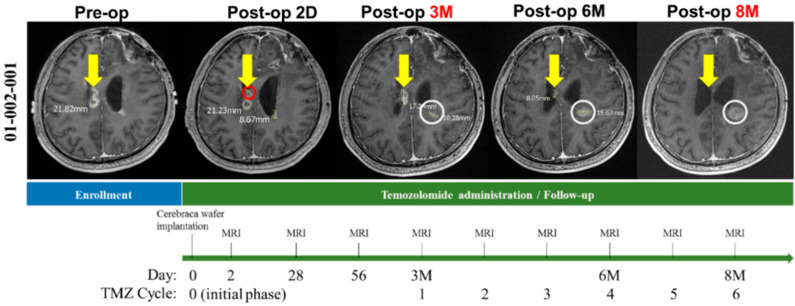
The clinical outcomes on Cerebraca wafer implantation. Representative magnetic resonance imaging (MRI) scans that illustrate the therapeutic effects of Cerebraca wafer implantation after surgical resection with succeeding temozolomide (TMZ) administration. The yellow arrow indicates the original tumor site; the red circle, the Cerebraca wafer implantation site; and the white circle, a new brain tumor lesion.

**Figure 3 cancers-14-01051-f003:**
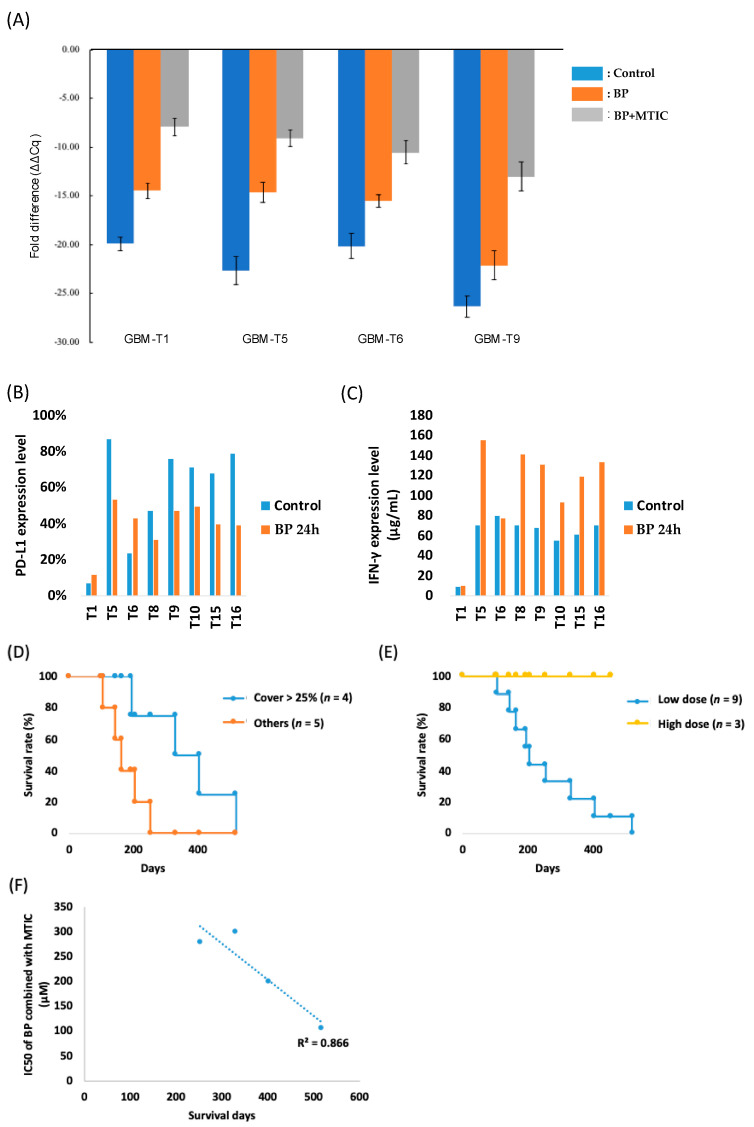
Molecular mechanism evaluation of the patient-derived glioblastoma (GBM) cell lines and analysis of the survival status in the phase I study. (**A**) Expression levels of MGMT were expressed as ΔΔCq which indicates the change in expression levels between drug treatment in the patient-derived glioblastoma (GBM) cells. Expression levels of (**B**) PD-L1 and (**C**) IFN-γ after BP treatment and co-culture with immune cells. Kaplan–Meier curve of overall survival of the different groups. (**D**) The overall survival rates of the low-dose (one to four wafers) and high-dose (six wafers) groups. (**E**) The overall survival rate in patients with a wafer coverage of more than 25% and others in the low-dose group. (**F**) The correlation between patient survival and the combination effects of BP and MTIC.

**Table 1 cancers-14-01051-t001:** Patient characteristics, adverse events (AEs, ≥grade 3) and serious adverse events (SAEs) reported and the rate overall survival (OS) and progression-free survival (PFS) of the participants in this phase I.

Variables	Cerebraca Wafer (75 mg API/300 mg Excipient) Treatment Groups		
	75–150 mg API *n* = 6	300 mg API *n* = 3	450 mg API *n* = 3
Age (years)			
mean (min-max)	51.7 (37.9–68.9)	40.3 (27.0–51.1)	55.7 (50.1–65.5)
Gender			
male:female	5:1	2:1	2:1
Time from diagnosis (months)			
Mean (min-max)	17.0 (4.5–24.1)	15.7 (9.1–25.7)	26.8 (10.4–43.8)
Recurrence			
First	4	3	2
Second	2	0	0
Third or more	0	0	1
Prior treatment lines (mean)	3.5	3.3	3.7
Prior bevacizumab (Yes:No)	1:5	1:2	0:3
Grade IV:Grade III			
At screening	4:2	3:0	2:1
At study entry	5:1	3:0	3:0
Resection rate			
>95%	2/6	1/3	2/3
>75%	4/6	1/3	2/3
≤75%	2/6	2/3	1/3
Biopsy	1/6	1/3	0/3
wafer coverage rate			
>25% (exclude 95% resection)	3/4	2/2	1/1
Molecular subtype			
Classical (EGFR)	3	0	2
Mesenchymal (NF1)	1	3	1
Neural (NEFL)	1	0	0
Proneural (IDH^R132H^)	1	0	0
MGMT promoter methylation			
Unmethylated:Methylated	5:1	3:0	2:1
KPS (mean, min-max)			
At screening	81.7 (73.1–90.3)	90.0 (90.0–90.0)	80.0 (71.3–88.7)
At Day 28 (±1 day)	73.3 (62.0–84.6)	86.7 (83.8–89.6)	70.0 (60.0–80.0)
QLQ-C30 (mean, min-max)			
Health status			
At screening	45.8 (36.4–55.2)	75.0 (70.8–79.2)	55.6 (34.6–76.6)
At Day 28 (±1 day)	59.7 (47.2–72.2)	55.6 (46.0–65.2)	61.1 (41.9–80.3)
Functional scales			
At screening	67.8 (52.4–83.2)	85.2 (80.1–90.3)	57.0 (39.7–74.3)
At Day 28 (±1 day)	61.5 (48.5–74.5)	80.7 (74.6–86.8)	46.7 (27.4–66.0)
Symptom scales			
At screening	14.5 (10.1–18.9)	08.5 (07.0–10.0)	29.9 (27.2–32.6)
At Day 28 (±1 day)	22.6 (17.7–27.5)	18.8 (12.2–25.4)	20.5 (13.4–27.6)
Steroid use (Yes:No:Unknown)			
At study entry	1:5:0	0:3:0	0:3:0
At Day 0–21	3:3:0	1:2:0	0:3:0
Event term	*n* (ratio)	*n* (ratio)	*n* (ratio)
≥Grade 3 AE	2/6	1/3	1/3
Liver function abnormal	1/6		
Wound complication	1/6		
Sepsis		1/3	
Lung infection			1/3
≥Grade 3 neurologic AE	0/6	1/3	1/3
CSF leakage			1/3
Hydrocephalus		1/3	
Incidence of SAE	1/6	2/3	3/3
Wound complication	1/6		
Spinal compression fracture			1/3
Sepsis		1/3	
Lung infection			1/3
CSF leakage			1/3
Hydrocephalus		1/3	
Survival status	*n* (ratio)	*n* (ratio)	*n* (ratio)
OS			
At 6M	5/6	1/3	3/3
At 9M	3/6	0/3	3/3
At 12M	2/6	0/3	3/3
PFS			
At 6M	2/6	1/3	3/3
At 9M	0/6	0/3	1/3
At 12M	0/6	0/3	1/3

**Table 2 cancers-14-01051-t002:** Stem cell characterization and the synergistic effects of BP and MTIC in patient-derived primary glioblastoma cell lines.

Sample ID	T1	T5	T6	T8	T9	T10	T15	T16
Cohort	Cohort I			Cohort II		Cohort III	Cohort IV	
CD133	0.29%	88.12%	94.26%	91.81%	92.40%	89.70%	84.28%	91.20%
SOX2	0.50%	93.20%	90.75%	89.87%	94.26%	92.45%	80.40%	89.62%
BCNU IC_50_	1200 μM	2000 μM	>2000 μM	2000 μM	>2000 μM	>2000 μM	1800 μM	1600 μM
BP IC_50_	300 μM	400 μM	420 μM	400 μM	600μM	410 μM	390 μM	405 μM
MGMT unmethylation	+	+	+	−	+	+	+	+
MTIC IC_50_	300 μM	700 μM	300 μM	400 μM	>800 μM	375 μM	500 μM	460 μM
BP + MTIC IC_50_	105 μM	200 μM	200 μM	300 μM	280 μM	250 μM	300 μM	250 μM
BP 8 h + MTIC IC_50_	100 μM	225 μM	155 μM	150 μM	275 μM	260 μM	200 μM	190 μM

## Data Availability

The data presented in this study are available in insert article.

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
