# Peer review of "Interstitial Control-Released Polymer Carrying a Targeting Small-Molecule Drug Reduces PD-L1 and MGMT Expression in Recurrent High-Grade Gliomas with TMZ Resistance"

_cancers, 2022, doi:10.3390/cancers14041051_

Round 1

Reviewer 1 Report

In the present research, the authors show that direct delivery of EF-001 through Cerebraca wafer in surgically resected glioma tissues shows synergistic effects with temozolomide and is active at low IC50. In addition, it was also observed to lower the PD-L1 in turn making T cell more active leading to higher secretion of INF gamma which has anti-tumor properties. I have several reservations. My comments are appended as below:

Major comments:

  1. MGMT methylation- share details on prognosis.
  2. Reference 5,6,7- share details on prognosis and statistical inference if available.
  3. Auhtors state that ‘The excipient of the Cerebraca wafer is a biodegradable polyanhydride CPPSA co-84 polymer (similar to that in the Gliadel wafer) that is safe and has been intracranially de-85 livered to animals and patients for more than 20 years.. ‘ Please explain the type of study and model system.
  4. Author’s seems to do primary culture, do authors perform authentication?
  5. IFN-γ ELISA assay-which samples were used as starting material?
  6. Primary culture: do authors remove fibroblast before culture?
  7. Cancer stem cell: do authors attempt to perform a sphere assay which provides more accurate information on stem cell components?
  8. Table 2: do authors use any controls without treatment, without which it's hard to judge the merit of treatment.
  9. Figure 3- annotate all panels with statistical inference.
  10. Figure 3 D, E- specify PFS or OS in the figure.
  11. Note the limitations of this study.

Minor comments:

  1. List the catalog number of all reagents used.
  2. RNA isolation and qRT PCR- how the purity of RNA was examined?

Author Response

Dear Reviewer:

It is with great pleasure that we resubmit our manuscript entitled "Interstitial control-released polymer carrying a targeting small-molecule drug reduces    PD-L1 and MGMT expressions in recurrent high-grade gliomas with TMZ resistance" for your further consideration. We appreciate the time and effort that you have decided to providing  insightful feedback to strengthen our manuscript. We have incorporated changes that reflect the suggestions you have graciously provided. To facilitate your review of our revision, the following is a point-by-point response to your questions and comments. We hope that our revision and the responses we provided below can satisfactorily address all the issues and concerns you have noted.

Comments from Reviewer and the responses:

Reviewer#1

Comments and Suggestions for Authors

In the present research, the authors show that direct delivery of EF-001 through Cerebraca wafer in surgically resected glioma tissues shows synergistic effects with temozolomide and is active at low IC50. In addition, it was also observed to lower the PD-L1 in turn making T cell more active leading to higher secretion of INF gamma which has anti-tumor properties. I have several reservations. My comments are appended as below:

Major comments:

  1. MGMT methylation- share details on prognosis.

[Reply]

Thank you for the suggestion. The promoter methylation status of MGMT has been reported to be associated with patient survival. Around 95% of the GBM patients that survived longer than 30 months after treatment are MGMT methylated; while only 36% of the control patients (survived shooter than 30 months) are MGMT methylated [1].

In this study, only two subjects were defined as MGMT promotor methylated. One subject receive two wafers with an OS of 10.8 months, the other subject received six wafers with an OS of more than 17.4 months. Although it is hard to tell if the MGMT promoter methylation status affect the patient survival due to the small sample size, we do show that the in primary culture from these subjects in this study, lower IC50 of EF-001 plus MTIC (which was correlated to the reduced expression levels of MGMT) is associated to longer patient survival (figure 3F).

    The contents in the first paragraph were incorporate in to the introduction part (Line 65 to Line 68) 

  1. Reference 5,6,7- share details on prognosis and statistical inference if available.

[Reply]

Thank you very much for the comment. The introduction for the expression of PD-1/PD-L1 to the prognosis and statistical inference was provided according to the following reference (Reference 5,6,7):

  • Chen, R.Q.; Liu, F.; Qiu, X.Y.; Chen, X.Q. The Prognostic and Therapeutic Value of PD-L1 in Glioma. Front Pharmacol 2018, 9, 1503
  • Wang, X.; Guo, G.; Guan, H.; Yu, Y.; Lu, J.; Yu, J. Challenges and potential of PD-1/PD-L1 checkpoint blockade immunotherapy for glioblastoma
  • Tomaszewski, W.; Sanchez-Perez, L.; Gajewski, T.F.; Sampson, J.H. Brain Tumor Microenvironment and Host State: Implications for Immunotherapy.

The correlations between PD-1/PD-L1 and prognosis in GBM are still uncertain. According to the Cancer Genome Atlas (TCGA) and Chinese Glioma Genome Atlas (CGGA) database, a high expression of both PD-L1and PD-1 negatively correlates with prognosis of patients (P = 0.0031 and P = 0.0253 respectively)[2, 3], however, no significant correlations was reported in some other study [4, 5].

The contents of the above paragraph were incorporate in to the introduction part (Line 80 to Line 84)

  1. Authors state that ‘The excipient of the Cerebraca wafer is a biodegradable polyanhydride CPPSA copolymer (similar to that in the Gliadel wafer) that is safe and has been intracranially delivered to animals and patients for more than 20 years. ‘ Please explain the type of study and model system.

[Reply]

Thank you for the comments, introduction about the excipient of Cerebraca Wafer has been revised in introduction (Line 95 to Line 115) and stated as follows: The excipient of the Cerebraca wafer is a biodegradable polyanhydride CPPSA copolymer (similar to that in the Gliadel wafer) that is safe and has been intracranially delivered to animals and patients for more than 20 years. Polyanhydrides are biodegradable polymers have been developed since 1980 [6]. The toxicology studies have been performed on the rat and monkey brain through the implantation route [7, 8]. In the rat study, 42 hemispheres of adult Sprague-Dawley rats’ brains received bilateral frontal lobe implantation of CPPSA. None of the animals showed any behavioral changes or neurological deficits[8]. In the monkey study, the group that received CPPSA brain implantation revealed no neurological or general deleterious effects[7]. The Gliadel wafer has been approved by FDA since 1996 as evidenced by the pivotal phase III clinical trial which included 120 GBM patients receiving Gliadel wafer and 120 patients receiving blank wafer (CPPSA only) as a placebo control)[9]. These studies provide the safety information of the excipient of Cerebraca Wafer.

  1. Author’s seems to do primary culture, do authors perform authentication?

[Reply]

Thanks a ton for your comments. The use of human brain tissue and peripheral blood samples was followed ethical and technical guidelines on the use of human samples for biomedical research purposes. Patient glioblastoma tissue and peripheral blood samples were collected at the Hualien Tzu Chi Hospital after informed patient consent under a protocol IRB-108-15A approved by the Hualien Tzu Chi Hospital Institutional Review Board.

  1. IFN-γ ELISA assay-which samples were used as starting material?

[Reply]

We are all gratitude for your comments. The starting material of IFN-γ ELISA assay was the collected culture medium from GBM patient-isolated cancer cells and the patient PBMCs co-culture. IFN-γ levels in the supernatant were determined with ELISA kits (Cat.430104, BioLegend) according to manufacturer’s instructions.

  1. Primary culture: do authors remove fibroblast before culture?

[Reply]

Thanks a million for the expert suggestion. The GBM primary cells were isolated using Papain Dissociation System (Cat. LK003150, Worthington Biochemical Corporation, Freehold, NJ, USA) that is a set of reagents intended for use in the neural cell isolation method of Huettner and Baughman (J. Neurosci., 6, 3044, 1986). However, the isolated GBM primary cells were stained with 1B10 (NB100-1845, Novus Biologicals), which is shown to react with human fibroblasts and cell lines, and FAP (BMS168, Thermo Fisher Scientific) that is highly expressed in activated fibroblasts. Both of 1B10 and FAP are not detected in isolated GBM primary cells by FACS analysis.

  1. Cancer stem cell: do authors attempt to perform a sphere assay which provides more accurate information on stem cell components?

[Reply]

Thanks for taking the time to review and give us so valuable comments. We perform the sphere assay of BCNU and EF-001 IC50 experiments on GBM attached cells and spheroids at the same time. The trends of these two cell types are identical. The IC50 of GBM spheroids are higher than attached cells.

  1. Table 2: do authors use any controls without treatment, without which it's hard to judge the merit of treatment.

[Reply]

Thank you for the comments. This study is a single-arm open-level phase I clinical trial based on the FDA approval whose primary endpoint aims to evaluate the maximum tolerated dose and dose limiting toxicities of Cerebraca Wafer. Although no placebo control arm was designed in this study, we tried to use historical controls of different strategy utilized on the treatment of recurrent GBM (similar to our inclusion criteria), such as Avastin and Gliadel wafer (Line 446 to Line 449).

  1. Figure 3- annotate all panels with statistical inference.

[Reply]

We do appreciate reviewer’s notification. The statistical inference has calculated and redrawed the figure 3B and figure 3C.

  1. Figure 3 D, E- specify PFS or OS in the figure.

[Reply]

Thank you for the comments. The figure 3 D, E are the analysis of patient overall survival, which is been revised in the updated manuscript.

  1. Note the limitations of this study.

[Reply]

Thank you for the comments. We have included the limitations of this study in the discussion section (Line 488 to Line 490) as follows: Although we showed an exciting result of Cerebraca Wafer treatment in this phase I study, further clinical trials that include more patients to achieve the effective sample size are required.

Minor comments:

  1. List the catalog number of all reagents used.

[Reply]

Thank you so much to remind us the incomplete part of all reagents. The catalog number of all reagents we used have listed in the updated manuscript.

  1. RNA isolation and qRT PCR- how the purity of RNA was examined?

[Reply]

Thanks for the expert comment. Total RNA of GBM primary cells was isolated with RNeasy Plus reagent (Cat. 74134, Qiagen) according to the manufacturer’s protocol. The total RNA extraction buffer was additional added RNase-free DNase I (Cat. 79254, Qiagen).

To estimate RNA purity, the ratio of the absorbance contributed by the nucleic acid to the absorbance of the contaminants is calculated. NanoDrop® spectrophotometer at OD260nm, OD280nm and OD230nm were measured. The ratio of OD260/OD280 are ranging from 1.98 to 2.05, OD260/OD230 ratio are from 1.82 to 1.96. In addition, we also perform the agarose gel electrophoresis to check RNA quality and purity. The agarose gel was stained with SYBRâ Green fluorescent dye, the separated fragments then can be visualized by excitation of the fluorescent dye bound to the nucleic acid. Whole gel imaging shows two bands correlated to 28S and 18S ribosomal RNA with the ratio of 2.2:1.

  1. Smrdel, U.; Popovic, M.; Zwitter, M.; Bostjancic, E.; Zupan, A.; Kovac, V.; Glavac, D.; Bokal, D.; Jerebic, J., Long-term survival in glioblastoma: methyl guanine methyl transferase (MGMT) promoter methylation as independent favourable prognostic factor. Radiol Oncol 2016, 50, (4), 394-401.
  2. Nduom, E. K.; Wei, J.; Yaghi, N. K.; Huang, N.; Kong, L. Y.; Gabrusiewicz, K.; Ling, X.; Zhou, S.; Ivan, C.; Chen, J. Q.; Burks, J. K.; Fuller, G. N.; Calin, G. A.; Conrad, C. A.; Creasy, C.; Ritthipichai, K.; Radvanyi, L.; Heimberger, A. B., PD-L1 expression and prognostic impact in glioblastoma. Neuro Oncol 2016, 18, (2), 195-205.
  3. Wang, Z.; Zhang, C. B.; Liu, X.; Wang, Z. L.; Sun, L. H.; Li, G. Z.; Liang, J. S.; Hu, H. M.; Liu, Y. W.; Zhang, W.; Jiang, T., Molecular and clinical characterization of PD-L1 expression at transcriptional level via 976 samples of brain glioma. OncoImmunology 2016, 5, (11).
  4. Berghoff, A. S.; Kiesel, B.; Widhalm, G.; Rajky, O.; Ricken, G.; Wöhrer, A.; Dieckmann, K.; Filipits, M.; Brandstetter, A.; Weller, M.; Kurscheid, S.; Hegi, M. E.; Zielinski, C. C.; Marosi, C.; Hainfellner, J. A.; Preusser, M.; Wick, W., Programmed death ligand 1 expression and tumor-infiltrating lymphocytes in glioblastoma. Neuro Oncol 2015, 17, (8), 1064-75.
  5. Zeng, J.; Zhang, X. K.; Chen, H. D.; Zhong, Z. H.; Wu, Q. L.; Lin, S. X., Expression of programmed cell death-ligand 1 and its correlation with clinical outcomes in gliomas. Oncotarget 2016, 7, (8), 8944-55.
  6. Su, Q.; Zhao, A.; Peng, H.; Zhou, S., Preparation and characterization of biodegradable electrospun polyanhydride nano/microfibers. J Nanosci Nanotechnol 2010, 10, (10), 6369-75.
  7. Brem, H.; Tamargo, R. J.; Olivi, A.; Pinn, M.; Weingart, J. D.; Wharam, M.; Epstein, J. I., Biodegradable polymers for controlled delivery of chemotherapy with and without radiation therapy in the monkey brain. J Neurosurg 1994, 80, (2), 283-90.
  8. Tamargo, R. J.; Epstein, J. I.; Reinhard, C. S.; Chasin, M.; Brem, H., Brain biocompatibility of a biodegradable, controlled-release polymer in rats. J Biomed Mater Res 1989, 23, (2), 253-66.
  9. Westphal, M.; Hilt, D. C.; Bortey, E.; Delavault, P.; Olivares, R.; Warnke, P. C.; Whittle, I. R.; Jaaskelainen, J.; Ram, Z., A phase 3 trial of local chemotherapy with biodegradable carmustine (BCNU) wafers (Gliadel wafers) in patients with primary malignant glioma. Neuro Oncol 2003, 5, (2), 79-88.
  10. Harn, H. J.; Lin, S. Z.; Lin, P. C.; Liu, C. Y.; Liu, P. Y.; Chang, L. F.; Yen, S. Y.; Hsieh, D. K.; Liu, F. C.; Tai, D. F.; Chiou, T. W., Local interstitial delivery of z-butylidenephthalide by polymer wafers against malignant human gliomas. Neuro Oncol 2011, 13, (6), 635-48.
  11. Suzuki, C.; Jacobsson, H.; Hatschek, T.; Torkzad, M. R.; Boden, K.; Eriksson-Alm, Y.; Berg, E.; Fujii, H.; Kubo, A.; Blomqvist, L., Radiologic measurements of tumor response to treatment: practical approaches and limitations. Radiographics 2008, 28, (2), 329-44.
  12. Lin, P. C.; Chen, Y. L.; Chiu, S. C.; Yu, Y. L.; Chen, S. P.; Chien, M. H.; Chen, K. Y.; Chang, W. L.; Lin, S. Z.; Chiou, T. W.; Harn, H. J., Orphan nuclear receptor, Nurr-77 was a possible target gene of butylidenephthalide chemotherapy on glioblastoma multiform brain tumor. Journal of neurochemistry 2008, 106, (3), 1017-1026.
  13. Lin, P. C.; Lin, S. Z.; Chen, Y. L.; Chang, J. S.; Ho, L. I.; Liu, P. Y.; Chang, L. F.; Harn, Y. C.; Chen, S. P.; Sun, L. Y.; Huang, P. C.; Chein, J. T.; Tsai, C. H.; Chou, C. W.; Harn, H. J.; Chiou, T. W., Butylidenephthalide suppresses human telomerase reverse transcriptase (TERT) in human glioblastomas. Ann Surg Oncol 2011, 18, (12), 3514-27.
  14. Huang, M. H.; Lin, S. Z.; Lin, P. C.; Chiou, T. W.; Harn, Y. W.; Ho, L. I.; Chan, T. M.; Chou, C. W.; Chuang, C. H.; Su, H. L.; Harn, H. J., Brain tumor senescence might be mediated by downregulation of S-phase kinase-associated protein 2 via butylidenephthalide leading to decreased cell viability. Tumor Biol 2014, 35, (5), 4875-4884.
  15. Delaney, C.; Garg, S. K.; Yung, R., Analysis of DNA Methylation by Pyrosequencing. Methods Mol Biol 2015, 1343, 249-64.
  16. Yen, S.Y.; Chen, S.R.; Hsieh, J.; Li, Y.S.; Chuang, S.E.; Chuang, H.M.; Huang, M.H.; Lin, S.Z.; Harn, H.J.; Chiou, T.W. Biodegradable interstitial release polymer loading a novel small molecule targeting Axl receptor tyrosine kinase and reducing brain tumour migration and invasion. Oncogene 2016, 35, 2156-2165, doi:10.1038/onc.2015.277.
  17. Yen, S.Y.; Chuang, H.M.; Huang, M.H.; Lin, S.Z.; Chiou, T.W.; Harn, H.J. n-Butylidenephthalide Regulated Tumor Stem Cell Genes EZH2/AXL and Reduced Its Migration and Invasion in Glioblastoma. Int J Mol Sci 2017, 18, doi:10.3390/ijms18020372.

Reviewer 2 Report

This work by Liu et al. reports on a polymer that was used therapeutically in glioblastoma patients. Cerebraca carries the pharmaceutically active agent EF-001 and is implanted intracranially as a wefer. Although only a small number of patients were treated (12 cases), the study nevertheless deserves attention, as the side effect profile and the therapeutic index were reported, which seem to be better than with BCNU wafers.

The following points must be taken into account in the revision:

  1. In Material and Methods I could not find any information on the origin of the wafers. Are the wafers commercially available? How can the work be reproduced?

  1. The active substance is EF-001. But nothing is said about EF-001 either in the introduction or in the discussion. What is EF-001? Which previous studies are available? Is EF-001 genotoxic? In the discussion it is only mentioned that EF-001 has a lower EC50 than BCNU. Reference 24 dates from 2006. From this I learned that this natural compound (or mixture of compounds?) induces apoptosis. What is the cytotoxic mechanism? Are there any more recent works on this? Please add this important information on EF-001 in the Discussion.

  1. In Results, 3.5. Molecular analysis of primary glioma cell lines was presented. However, no data were shown.

  1. Cerebraca was reported to have an impact on MGMT expression by upregulating promoter methylation and, therefore, downregulation of MGMT activity. Was MGMT promoter methylation determined by methylation-specific PCR? Please add reference. Furthermore, if MGMT promoter methylation is compared in primary tumors versus recurrences after Cerebraca therapy, was there a change in the promoter methylation status?

  1. How good are the in vitro data to be able to be transferred to the patient? Does the EF-001 IC50 in T1 to T16 correlate with the patient's response?

Author Response

Dear Reviewer:

It is with great pleasure that we resubmit our manuscript entitled "Interstitial control-released polymer carrying a targeting small-molecule drug reduces    PD-L1 and MGMT expressions in recurrent high-grade gliomas with TMZ resistance" for your further consideration. We appreciate the time and effort that you have decided to providing  insightful feedback to strengthen our manuscript. We have incorporated changes that reflect the suggestions you have graciously provided. To facilitate your review of our revision, the following is a point-by-point response to your questions and comments. We hope that our revision and the responses we provided below can satisfactorily address all the issues and concerns you have noted.

Comments from Reviewer and the responses:

Reviewer#2

In this article, Liu C et al reported a single-arm phase I trial of EF-001-loaded Cerebraca wafer to treat Glioblastoma patients. 3 patients were treated with 1/2/4/6 wafers during surgery to enhance efficacy of TMZ administration. The articles presented quite a thorough characterization in terms of disease progression, adverse events and molecular study from harvested tumor tissues.

Overall, I believe this article is of significant quality for Cancers, and suggest the acceptance of this article following several revision/clarifications below:

  1. Suggest the authors to include a summary of the inclusion/exclusion criteria in the manuscript

[Reply]

This suggestion is appreciated. The summary of inclusion and exclusion criteria has been included in the materials & methods section, as indicated in Line 156 to Line 193.

  1. Intro to physical properties of Cerebraca wafer & its excipient would be helpful (size, degradation profile, etc.). Related to this, how does the Cerebraca wafer presents a significantly greater diffusion distance than Gliadel wafer?

[Reply]

Thank you for your comments. The introduction section has been revised to include the information of Cerebraca Wafer (Line 106 to Line 115). Cerebraca Wafer is a product of Everfront biotech inc. which was composed of the API EF-001 and biodegradable excipient. Each 300mg Cerebraca wafer was manufactured by 25% EF-001 wafer powder. The manufacturing procedures of Cerebraca Wafer complied with Good Manufacturing Practice (GMP) set forth by the US FDA and Ministry of Health and Welfare, Taiwan. The in vitro release profile of Cerebraca Wafer indicates that the drug can be slow-released for at least 21 days[10]. Unlike the Gliadel wafer that only contains 3.8% of API, Cerebraca Wafer contains 25% of the API, which provides a higher local concentration after implantation. This high local concentration allowed the drug to achieve a greater diffusion distance. 

  1. How was wafer coverage determined?

[Reply]

Thank you for your comments. The materials and methods section has been revised to include the method for determining the wafer coverage (Line 197 to 202). The longest diameter and the longest perpendicular diameter obtained from the MRI image are multiplied to estimate the tumor area according to the WHO guideline [11]. A similar method (longest diameter and the longest perpendicular diameter are multiplied) was used to calculate the surface area of a wafer. The wafer coverage is determined by the wafer surface area / the tumor area.

  1. Is there data pertaining to how sustained is the PL-L1 reduction by EF-001?

[Reply]

Thank you for your comments. The EF-001 concentration in the peripheral blood was monitored in the study to confirm the slow-release characteristic of Cerebraca Wafer. We found that EF-001 can be detected within 21 days after the Wafer implantation, indicating the drug release can sustain for at least 21 days. This may provide evidence that PD-L1 expression in the brain tumor cell can be suppressed for at least 21 days.

  1. Please clarify for table 2: EF-001 + MTIC means co-application whereas EF-001 8h + MTIC means 8h pretreatment of EF-001?

[Reply]

Thank you for the comments. Yes, EF-001 + MTIC means co-application, EF-001 8h + MTIC means 8h pretreatment of EF-001. Our previous studies demonstrated that EF-001 has a combination effect with TMZ and reduces MGMT expression to overcome TMZ drug resistance [16, 17]. Therefore, we would like to understand if there is different effect of co-treatment and sequential treatment of the cytotoxicity on GBM stem cells. The results reveal that IC50 of EF-001 pre-treatment followed by treating with active form of TMZ (MTIC) is lower than co-treatment.   

  1. Please check & correct figure caption to match the labeling in Figure 3

[Reply]

The figure 3 caption has been checked and corrected to match the labelling.

  1. Please check & correct several awkward phrasing: "are still survived", "generalizability", "much more improved in compare"

[Reply]

Think you for your notice. The term "are still survived" was revised to "not expired"; "to increase the generalizability of the findings" was revised to "to increase the confidence of the findings"; "much more improved in compare" was revised to "better than".

  1. Smrdel, U.; Popovic, M.; Zwitter, M.; Bostjancic, E.; Zupan, A.; Kovac, V.; Glavac, D.; Bokal, D.; Jerebic, J., Long-term survival in glioblastoma: methyl guanine methyl transferase (MGMT) promoter methylation as independent favourable prognostic factor. Radiol Oncol 2016, 50, (4), 394-401.
  2. Nduom, E. K.; Wei, J.; Yaghi, N. K.; Huang, N.; Kong, L. Y.; Gabrusiewicz, K.; Ling, X.; Zhou, S.; Ivan, C.; Chen, J. Q.; Burks, J. K.; Fuller, G. N.; Calin, G. A.; Conrad, C. A.; Creasy, C.; Ritthipichai, K.; Radvanyi, L.; Heimberger, A. B., PD-L1 expression and prognostic impact in glioblastoma. Neuro Oncol 2016, 18, (2), 195-205.
  3. Wang, Z.; Zhang, C. B.; Liu, X.; Wang, Z. L.; Sun, L. H.; Li, G. Z.; Liang, J. S.; Hu, H. M.; Liu, Y. W.; Zhang, W.; Jiang, T., Molecular and clinical characterization of PD-L1 expression at transcriptional level via 976 samples of brain glioma. OncoImmunology 2016, 5, (11).
  4. Berghoff, A. S.; Kiesel, B.; Widhalm, G.; Rajky, O.; Ricken, G.; Wöhrer, A.; Dieckmann, K.; Filipits, M.; Brandstetter, A.; Weller, M.; Kurscheid, S.; Hegi, M. E.; Zielinski, C. C.; Marosi, C.; Hainfellner, J. A.; Preusser, M.; Wick, W., Programmed death ligand 1 expression and tumor-infiltrating lymphocytes in glioblastoma. Neuro Oncol 2015, 17, (8), 1064-75.
  5. Zeng, J.; Zhang, X. K.; Chen, H. D.; Zhong, Z. H.; Wu, Q. L.; Lin, S. X., Expression of programmed cell death-ligand 1 and its correlation with clinical outcomes in gliomas. Oncotarget 2016, 7, (8), 8944-55.
  6. Su, Q.; Zhao, A.; Peng, H.; Zhou, S., Preparation and characterization of biodegradable electrospun polyanhydride nano/microfibers. J Nanosci Nanotechnol 2010, 10, (10), 6369-75.
  7. Brem, H.; Tamargo, R. J.; Olivi, A.; Pinn, M.; Weingart, J. D.; Wharam, M.; Epstein, J. I., Biodegradable polymers for controlled delivery of chemotherapy with and without radiation therapy in the monkey brain. J Neurosurg 1994, 80, (2), 283-90.
  8. Tamargo, R. J.; Epstein, J. I.; Reinhard, C. S.; Chasin, M.; Brem, H., Brain biocompatibility of a biodegradable, controlled-release polymer in rats. J Biomed Mater Res 1989, 23, (2), 253-66.
  9. Westphal, M.; Hilt, D. C.; Bortey, E.; Delavault, P.; Olivares, R.; Warnke, P. C.; Whittle, I. R.; Jaaskelainen, J.; Ram, Z., A phase 3 trial of local chemotherapy with biodegradable carmustine (BCNU) wafers (Gliadel wafers) in patients with primary malignant glioma. Neuro Oncol 2003, 5, (2), 79-88.
  10. Harn, H. J.; Lin, S. Z.; Lin, P. C.; Liu, C. Y.; Liu, P. Y.; Chang, L. F.; Yen, S. Y.; Hsieh, D. K.; Liu, F. C.; Tai, D. F.; Chiou, T. W., Local interstitial delivery of z-butylidenephthalide by polymer wafers against malignant human gliomas. Neuro Oncol 2011, 13, (6), 635-48.
  11. Suzuki, C.; Jacobsson, H.; Hatschek, T.; Torkzad, M. R.; Boden, K.; Eriksson-Alm, Y.; Berg, E.; Fujii, H.; Kubo, A.; Blomqvist, L., Radiologic measurements of tumor response to treatment: practical approaches and limitations. Radiographics 2008, 28, (2), 329-44.
  12. Lin, P. C.; Chen, Y. L.; Chiu, S. C.; Yu, Y. L.; Chen, S. P.; Chien, M. H.; Chen, K. Y.; Chang, W. L.; Lin, S. Z.; Chiou, T. W.; Harn, H. J., Orphan nuclear receptor, Nurr-77 was a possible target gene of butylidenephthalide chemotherapy on glioblastoma multiform brain tumor. Journal of neurochemistry 2008, 106, (3), 1017-1026.
  13. Lin, P. C.; Lin, S. Z.; Chen, Y. L.; Chang, J. S.; Ho, L. I.; Liu, P. Y.; Chang, L. F.; Harn, Y. C.; Chen, S. P.; Sun, L. Y.; Huang, P. C.; Chein, J. T.; Tsai, C. H.; Chou, C. W.; Harn, H. J.; Chiou, T. W., Butylidenephthalide suppresses human telomerase reverse transcriptase (TERT) in human glioblastomas. Ann Surg Oncol 2011, 18, (12), 3514-27.
  14. Huang, M. H.; Lin, S. Z.; Lin, P. C.; Chiou, T. W.; Harn, Y. W.; Ho, L. I.; Chan, T. M.; Chou, C. W.; Chuang, C. H.; Su, H. L.; Harn, H. J., Brain tumor senescence might be mediated by downregulation of S-phase kinase-associated protein 2 via butylidenephthalide leading to decreased cell viability. Tumor Biol 2014, 35, (5), 4875-4884.
  15. Delaney, C.; Garg, S. K.; Yung, R., Analysis of DNA Methylation by Pyrosequencing. Methods Mol Biol 2015, 1343, 249-64.
  16. Yen, S.Y.; Chen, S.R.; Hsieh, J.; Li, Y.S.; Chuang, S.E.; Chuang, H.M.; Huang, M.H.; Lin, S.Z.; Harn, H.J.; Chiou, T.W. Biodegradable interstitial release polymer loading a novel small molecule targeting Axl receptor tyrosine kinase and reducing brain tumour migration and invasion. Oncogene 2016, 35, 2156-2165, doi:10.1038/onc.2015.277.
  17. Yen, S.Y.; Chuang, H.M.; Huang, M.H.; Lin, S.Z.; Chiou, T.W.; Harn, H.J. n-Butylidenephthalide Regulated Tumor Stem Cell Genes EZH2/AXL and Reduced Its Migration and Invasion in Glioblastoma. Int J Mol Sci 2017, 18, doi:10.3390/ijms18020372.

Reviewer 3 Report

In this article, Liu C et al reported a single-arm phase I trial of EF-001-loaded Cerebraca wafer to treat Glioblastoma patients. 3 patients were treated with 1/2/4/6 wafers during surgery to enhance efficacy of TMZ administration. The articles presented quite a thorough characterization in terms of disease progression, adverse events and molecular study from harvested tumor tissues. 
Overall, I believe this article is of significant quality for Cancers, and suggest the acceptance of this article following several revision/clarifications below:

  • Suggest the authors to include a summary of the inclusion/exclusion criteria in the manuscript
  • Intro to physical properties of Cerebraca wafer & its excipient would be helpful (size, degradation profile, etc.). Related to this, how does the Cerebraca wafer presents a significantly greater diffusion distance than Gliadel wafer?
  • How was wafer coverage determined?
  • Is there data pertaining to how sustained is the PL-L1 reduction by EF-001?
  • Please clarify for table 2: EF-001 + MTIC means co-application whereas EF-001 8h + MTIC means 8h pretreatment of EF-001?
  • Please check & correct figure caption to match the labeling in Figure 3
  • Please check & correct several awkward phrasing: "are still survived", "generalizability", "much more improved in compare"

Author Response

Dear Reviewer:

It is with great pleasure that we resubmit our manuscript entitled "Interstitial control-released polymer carrying a targeting small-molecule drug reduces    PD-L1 and MGMT expressions in recurrent high-grade gliomas with TMZ resistance" for your further consideration. We appreciate the time and effort that you have decided to providing  insightful feedback to strengthen our manuscript. We have incorporated changes that reflect the suggestions you have graciously provided. To facilitate your review of our revision, the following is a point-by-point response to your questions and comments. We hope that our revision and the responses we provided below can satisfactorily address all the issues and concerns you have noted.

Comments from Reviewer and the responses:

Reviewer #3

This work by Liu et al. reports on a polymer that was used therapeutically in glioblastoma patients. Cerebraca carries the pharmaceutically active agent EF-001 and is implanted intracranially as a wefer. Although only a small number of patients were treated (12 cases), the study nevertheless deserves attention, as the side effect profile and the therapeutic index were reported, which seem to be better than with BCNU wafers.

The following points must be taken into account in the revision:

  1. In Material and Methods I could not find any information on the origin of the wafers. Are the wafers commercially available? How can the work be reproduced?

 [Reply]

Thank you for the notice. An introduction of Cerebraca Wafer was added into the introduction section (Line 106 to Line 115). Cerebraca Wafer is a product of Everfront biotech inc. which was composed of the API EF-001 and biodegradable excipient. Each 300mg Cerebraca wafer was manufactured by 25% EF-001 wafer powder. The manufacturing procedures of Cerebraca Wafer complied with Good Manufacturing Practice (GMP) set forth by the US FDA and Ministry of Health and Welfare, Taiwan.

  1. The active substance is EF-001. But nothing is said about EF-001 either in the introduction or in the discussion. What is EF-001? Which previous studies are available? Is EF-001 genotoxic? In the discussion it is only mentioned that EF-001 has a lower EC50 than BCNU. Reference 24 dates from 2006. From this I learned that this natural compound (or mixture of compounds?) induces apoptosis. What is the cytotoxic mechanism? Are there any more recent works on this? Please add this important information on EF-001 in the Discussion.

[Reply]

Thank you for the comments. The therapeutic mechanism of action of EF-001 was revised and mentioned in the introduction part (Line 117 to Line 131). EF-001 exerted suppressive effects on tumor cell proliferation in human, rat and mice GBM cell lines via down-regulating the cell-cycle regulators and increasing the apoptosis-associated genes and proteins [12-14]. Moreover, in vivo results indicated that the subcutaneous injection of (Z)-BP in xenograft mouse models not only suppressed human GBM tumor growth but also prolonged survival rates [13, 14].

  1. In Results, 3.5. Molecular analysis of primary glioma cell lines was presented. However, no data were shown.

[Reply]

Appreciate for your comment. The molecular analyses of primary glioma cell were performed by TMZ resistance as evaluating by MGMT pyrosequencing, cancer stem cell marker CD133 and SOX2 identification, half-maximal inhibitory concentrations (IC50) of EF-001, BCNU and MTIC (the active form of TMZ). In addition, the PD-L1 expression and immune response of GBM/PBMC co-culture detection by IFN-g secretion. Data were presented in Table 2 and Figure 3. 

  1. Cerebraca was reported to have an impact on MGMT expression by upregulating promoter methylation and, therefore, downregulation of MGMT activity. Was MGMT promoter methylation determined by methylation-specific PCR? Please add reference. Furthermore, if MGMT promoter methylation is compared in primary tumors versus recurrences after Cerebraca therapy, was there a change in the promoter methylation status?

[Reply]

Thank you for the comments. The methods to determine MGMT promotor methylation was added in the materials and methods section (Line 301 to Line 306). MGMT promoter methylation status was evaluated by Pyrosequencing[15]. In brief, the genomic DNA from patient tumor tissue was extracted, then PCR was used to amplify the MGMT promoter region. The ssDNA isolated from PCR product was used as a template of Pyrosequencing. The pyrosequencing analysis was performed by QIAGEN PyroMark Q24. In the study, there is only one case that has a chance to perform Pyrosequencing in primary tumors versus recurrences after Cerebraca Wafer implantation. It was found that methylation status increased from 4.2% (in the primary tumor) to 15.4% (after Cerebraca Wafer implantation, then recurrent). This result together with the down-regulated MGMT expression levels after EF-001 treatment in the patient-derived cell lines, suggests that after implantation of Cerebraca Wafer to the patient’s brain, the methylation status of MGMT promoter is altered by the API EF-001.

  1. How good are the in vitro data to be able to be transferred to the patient? Does the EF-001 IC50 in T1 to T16 correlate with the patient's response?

[Reply]

Thank you for the comments. This is an add-on study; both Cerebraca Wafer and chemo adjuvant TMZ could have benefits for the patients recruited. Thus, the IC50 combine with the EF-001 and MTIC (the active form of TMZ) is determined. Although it needs further study to support our findings, as shown in figure 3F, lower IC50 of EF-001 combined with MTIC, longer patient survival is achieved.

  1. Smrdel, U.; Popovic, M.; Zwitter, M.; Bostjancic, E.; Zupan, A.; Kovac, V.; Glavac, D.; Bokal, D.; Jerebic, J., Long-term survival in glioblastoma: methyl guanine methyl transferase (MGMT) promoter methylation as independent favourable prognostic factor. Radiol Oncol 2016, 50, (4), 394-401.
  2. Nduom, E. K.; Wei, J.; Yaghi, N. K.; Huang, N.; Kong, L. Y.; Gabrusiewicz, K.; Ling, X.; Zhou, S.; Ivan, C.; Chen, J. Q.; Burks, J. K.; Fuller, G. N.; Calin, G. A.; Conrad, C. A.; Creasy, C.; Ritthipichai, K.; Radvanyi, L.; Heimberger, A. B., PD-L1 expression and prognostic impact in glioblastoma. Neuro Oncol 2016, 18, (2), 195-205.
  3. Wang, Z.; Zhang, C. B.; Liu, X.; Wang, Z. L.; Sun, L. H.; Li, G. Z.; Liang, J. S.; Hu, H. M.; Liu, Y. W.; Zhang, W.; Jiang, T., Molecular and clinical characterization of PD-L1 expression at transcriptional level via 976 samples of brain glioma. OncoImmunology 2016, 5, (11).
  4. Berghoff, A. S.; Kiesel, B.; Widhalm, G.; Rajky, O.; Ricken, G.; Wöhrer, A.; Dieckmann, K.; Filipits, M.; Brandstetter, A.; Weller, M.; Kurscheid, S.; Hegi, M. E.; Zielinski, C. C.; Marosi, C.; Hainfellner, J. A.; Preusser, M.; Wick, W., Programmed death ligand 1 expression and tumor-infiltrating lymphocytes in glioblastoma. Neuro Oncol 2015, 17, (8), 1064-75.
  5. Zeng, J.; Zhang, X. K.; Chen, H. D.; Zhong, Z. H.; Wu, Q. L.; Lin, S. X., Expression of programmed cell death-ligand 1 and its correlation with clinical outcomes in gliomas. Oncotarget 2016, 7, (8), 8944-55.
  6. Su, Q.; Zhao, A.; Peng, H.; Zhou, S., Preparation and characterization of biodegradable electrospun polyanhydride nano/microfibers. J Nanosci Nanotechnol 2010, 10, (10), 6369-75.
  7. Brem, H.; Tamargo, R. J.; Olivi, A.; Pinn, M.; Weingart, J. D.; Wharam, M.; Epstein, J. I., Biodegradable polymers for controlled delivery of chemotherapy with and without radiation therapy in the monkey brain. J Neurosurg 1994, 80, (2), 283-90.
  8. Tamargo, R. J.; Epstein, J. I.; Reinhard, C. S.; Chasin, M.; Brem, H., Brain biocompatibility of a biodegradable, controlled-release polymer in rats. J Biomed Mater Res 1989, 23, (2), 253-66.
  9. Westphal, M.; Hilt, D. C.; Bortey, E.; Delavault, P.; Olivares, R.; Warnke, P. C.; Whittle, I. R.; Jaaskelainen, J.; Ram, Z., A phase 3 trial of local chemotherapy with biodegradable carmustine (BCNU) wafers (Gliadel wafers) in patients with primary malignant glioma. Neuro Oncol 2003, 5, (2), 79-88.
  10. Harn, H. J.; Lin, S. Z.; Lin, P. C.; Liu, C. Y.; Liu, P. Y.; Chang, L. F.; Yen, S. Y.; Hsieh, D. K.; Liu, F. C.; Tai, D. F.; Chiou, T. W., Local interstitial delivery of z-butylidenephthalide by polymer wafers against malignant human gliomas. Neuro Oncol 2011, 13, (6), 635-48.
  11. Suzuki, C.; Jacobsson, H.; Hatschek, T.; Torkzad, M. R.; Boden, K.; Eriksson-Alm, Y.; Berg, E.; Fujii, H.; Kubo, A.; Blomqvist, L., Radiologic measurements of tumor response to treatment: practical approaches and limitations. Radiographics 2008, 28, (2), 329-44.
  12. Lin, P. C.; Chen, Y. L.; Chiu, S. C.; Yu, Y. L.; Chen, S. P.; Chien, M. H.; Chen, K. Y.; Chang, W. L.; Lin, S. Z.; Chiou, T. W.; Harn, H. J., Orphan nuclear receptor, Nurr-77 was a possible target gene of butylidenephthalide chemotherapy on glioblastoma multiform brain tumor. Journal of neurochemistry 2008, 106, (3), 1017-1026.
  13. Lin, P. C.; Lin, S. Z.; Chen, Y. L.; Chang, J. S.; Ho, L. I.; Liu, P. Y.; Chang, L. F.; Harn, Y. C.; Chen, S. P.; Sun, L. Y.; Huang, P. C.; Chein, J. T.; Tsai, C. H.; Chou, C. W.; Harn, H. J.; Chiou, T. W., Butylidenephthalide suppresses human telomerase reverse transcriptase (TERT) in human glioblastomas. Ann Surg Oncol 2011, 18, (12), 3514-27.
  14. Huang, M. H.; Lin, S. Z.; Lin, P. C.; Chiou, T. W.; Harn, Y. W.; Ho, L. I.; Chan, T. M.; Chou, C. W.; Chuang, C. H.; Su, H. L.; Harn, H. J., Brain tumor senescence might be mediated by downregulation of S-phase kinase-associated protein 2 via butylidenephthalide leading to decreased cell viability. Tumor Biol 2014, 35, (5), 4875-4884.
  15. Delaney, C.; Garg, S. K.; Yung, R., Analysis of DNA Methylation by Pyrosequencing. Methods Mol Biol 2015, 1343, 249-64.
  16. Yen, S.Y.; Chen, S.R.; Hsieh, J.; Li, Y.S.; Chuang, S.E.; Chuang, H.M.; Huang, M.H.; Lin, S.Z.; Harn, H.J.; Chiou, T.W. Biodegradable interstitial release polymer loading a novel small molecule targeting Axl receptor tyrosine kinase and reducing brain tumour migration and invasion. Oncogene 2016, 35, 2156-2165, doi:10.1038/onc.2015.277.
  17. Yen, S.Y.; Chuang, H.M.; Huang, M.H.; Lin, S.Z.; Chiou, T.W.; Harn, H.J. n-Butylidenephthalide Regulated Tumor Stem Cell Genes EZH2/AXL and Reduced Its Migration and Invasion in Glioblastoma. Int J Mol Sci 2017, 18, doi:10.3390/ijms18020372.

Round 2

Reviewer 1 Report

I congratulate the author for providing the modifications, with that the manuscript is closer to publishing. I however recommend taking care of a few minor points:

  1. Point 4- authors seem misunderstood my point. By authentication, I mean STR sequencing.

2. Point 7- please indicate the figure where the details are incorporated. 

Author Response

Dear Reviewer:

It is with great pleasure that we resubmit our manuscript entitled "Interstitial control-released polymer carrying a targeting small-molecule drug reduces PD-L1 and MGMT expressions in recurrent high-grade gliomas with TMZ resistance" for your further consideration. We appreciate the time and effort that you have decided to providing insightful feedback to strengthen our manuscript. We have incorporated changes that reflect the suggestions you have graciously provided. To facilitate your review of our revision, the following is a point-by-point response to your questions and comments. We hope that our revision and the responses we provided below can satisfactorily address all the issues and concerns you have noted.

Reviewer 2 Report

The revision is significantly improved. However, an explanation of what EF-001 actually represents is still missing. The discussion only states that it is a novel small molecule drug. It is apparently very toxic, more toxic than BCNU (lower IC50). BCNU is a genotoxic anticancer drug. Is EF-001 also genotoxic? What is the pleiotropic effect based on?

I could only find the reference 38 published in 2006. In this paper n-butylidenephthalide (BP) was reported, a compound purchased from a company in UK. Is EF-001 identic to n-butylidenephthalide? Was n-butylidenephthalide renamed? Is EF-001 commercially available?

I consider this information essential if someone wants to reproduce the data.

Please add a paragraph on n-butylidenephthalide to the discussion.

Author Response

Dear Reviewer:

We have revised our manuscript entitled "Interstitial control-released polymer carrying a targeting small-molecule drug reduces PD-L1 and MGMT expressions in recurrent high-grade gliomas with TMZ resistance" for your consideration. Changes that reflect the suggestions you have graciously provided have been incorporated in this revised manuscript with track-changes. Your constructive critiques that help improve this manuscript are greatly appreciated.

Comments from Reviewer and the responses:

Reviewer#2

Comments and Suggestions for Authors

The revision is significantly improved. However, an explanation of what EF-001 actually represents is still missing. The discussion only states that it is a novel small molecule drug. It is apparently very toxic, more toxic than BCNU (lower IC50). BCNU is a genotoxic anticancer drug. Is EF-001 also genotoxic? What is the pleiotropic effect based on?

I could only find the reference 38 published in 2006. In this paper n-butylidenephthalide (BP) was reported, a compound purchased from a company in UK. Is EF-001 identic to n-butylidenephthalide? Was n-butylidenephthalide renamed? Is EF-001 commercially available?

I consider this information essential if someone wants to reproduce the data.

Please add a paragraph on n-butylidenephthalide to the discussion.

Point-to-point responses are described as follows:

  1. An explanation of what EF-001 actually represents is still missing. Is EF-001 identic to n-butylidenephthalide? Was n-butylidenephthalide renamed? Is EF-001 commercially available?

I consider this information essential if someone wants to reproduce the data.

Please add a paragraph on n-butylidenephthalide to the discussion.

[Reply]

Thank you very much for your comment. EF-001 is (Z)-n-butylidenephthalide (BP) which is the Z form geometric isomer of n-butylidenephthalide. In the commercially available n-butylidenephthalide, usually more than 85% are BP. To meet the requirement of regulatory authority, clinical-grade BP (with purity no less than 99.5%), named as EF001, was manufactured by Everfront Biotech company to perform the non-clinical safety studies, the efficacy experiments and the clinical trials. Therefore, EF-001 (BP) is not commercially available yet. The production process of clinical-grade BP complied with GMP (Good Manufacturing Practice), and the full CMC (Chemistry, Manufacturing, and Control) section, as a part of the IND (Investigational New Drug) submission package, has been reviewed and approved by the US FDA and TFDA (the Ministry of Health and Welfare, Taiwan).

In order to avoid the confusion, we explained the relationship between n-butylidenephthalide and BP in line 518-521 and replaced the EF-001 with BP in the manuscript. Based on your kind suggestions, we have already added one paragraph about n-butylidenephthalide ant its anticancer effects to the discussion (line 503 to 523)

  1. It is apparently very toxic, more toxic than BCNU (lower IC50). BCNU is a genotoxic anticancer drug.

[Reply]

    Thank you very much for your comment. EF-001 is not more toxic than BCNU based on the following findings.

Although EF-001 (BP) revealed lower IC50 than BCNU toward brain tumor primary cells (table 2), it has been demonstrated that BP only targeted cancer cells and revealed low cytotoxicity toward normal cells such as human bone marrow cells or mouse fibroblasts [1].

 Cerebraca wafer has higher therapeutic index than BCNU wafer. According to their MSDS (W333301, sigma-aldrich; C0400, sigma-aldrich), oral LD50 of n-butylidenephthalide (usually more than 85% are BP) and BCNU in rat are 1,850 mg/kg and 20 mg/kg, which can be converted to 471 mg BP/rat and 6 mg BCNU/rat respectively. We have previously demonstrated that 3 mg/rat of BP or BCNU containing wafer can suppress the RG2 brain tumor volume in more than 50% of the experimental rats [2], which can be set forth as ED50. The therapeutic index as indicated by LD50/ED50 of Cerebraca wafer and BCNU wafer are approximately 157 and 2 respectively, suggesting that Cerebraca Wafer is safer than BCNU wafer.

Hematological toxicity, thromboembolism, intracranial hemorrhage, bone marrow suppression, leukopenia, thrombocytopenia, anemia, and fatigue were reported to be the adverse events of recurrent GBM patients treated with BCNU through oral administration [3,4]. Previous studies on Gliadel wafer (BCNU wafer) implantation revealed local complications such as brain edema and hemorrhage [5-7]. However, there were no Cerebraca Wafer-related blood and lymphatic system disorders, cardiac disorders, hepatobiliary disorders, and nervous system disorders observed in this clinical trial. Due to no dose-limiting toxicity (DLT) reported in this phase I trial, future clinical trials conducting higher dose levels of Cerebraca Wafer might be performed.

  These information has been added to the discussion section (Line 447-458).

  1. BCNU is a genotoxic anticancer drug. Is EF-001 also genotoxic?

EF-001 is not genotoxic. In our pre-IND studies, the genotoxicity was evaluated through the AMES test, chromosomal aberration test, and micronucleus test. No genotoxicity was found under the tested conditions in these studies.

  1. What is the pleiotropic effect based on?

The pleiotropic effect of EF-001 might be due to the activation of the AMPK pathway [8], its down-stream AXL receptor [9], and the DNMT regulation [10]. It has been added in the discussion section (line 521 to 523).

  1. Liu, P.Y.; Sheu, J.J.; Lin, P.C.; Lin, C.T.; Liu, Y.J.; Ho, L.I.; Chang, L.F.; Wu, W.C.; Chen, S.R.; Chen, J.; et al. Expression of Nur77 induced by an n-butylidenephthalide derivative promotes apoptosis and inhibits cell growth in oral squamous cell carcinoma. Invest New Drugs 2012, 30, 79-89, doi:10.1007/s10637-010-9518-z.
  2. Harn, H.J.; Lin, S.Z.; Lin, P.C.; Liu, C.Y.; Liu, P.Y.; Chang, L.F.; Yen, S.Y.; Hsieh, D.K.; Liu, F.C.; Tai, D.F.; et al. Local interstitial delivery of z-butylidenephthalide by polymer wafers against malignant human gliomas. Neuro Oncol 2011, 13, 635-648, doi:10.1093/neuonc/nor021.
  3. Jungk, C.; Chatziaslanidou, D.; Ahmadi, R.; Capper, D.; Bermejo, J.L.; Exner, J.; von Deimling, A.; Herold-Mende, C.; Unterberg, A. Chemotherapy with BCNU in recurrent glioma: Analysis of clinical outcome and side effects in chemotherapy-naive patients. BMC Cancer 2016, 16, 81, doi:10.1186/s12885-016-2131-6.
  4. Reithmeier, T.; Graf, E.; Piroth, T.; Trippel, M.; Pinsker, M.O.; Nikkhah, G. BCNU for recurrent glioblastoma multiforme: efficacy, toxicity and prognostic factors. BMC Cancer 2010, 10, 30, doi:10.1186/1471-2407-10-30.
  5. Kleinberg, L.R.; Weingart, J.; Burger, P.; Carson, K.; Grossman, S.A.; Li, K.; Olivi, A.; Wharam, M.D.; Brem, H. Clinical course and pathologic findings after Gliadel and radiotherapy for newly diagnosed malignant glioma: implications for patient management. Cancer Invest 2004, 22, 1-9, doi:10.1081/cnv-120027575.
  6. Weber, E.L.; Goebel, E.A. Cerebral edema associated with Gliadel wafers: two case studies. Neuro Oncol 2005, 7, 84-89, doi:10.1215/S1152851704000614.
  7. Kuramitsu, S.; Motomura, K.; Natsume, A.; Wakabayashi, T. Double-edged Sword in the Placement of Carmustine (BCNU) Wafers along the Eloquent Area: A Case Report. NMC Case Rep J 2015, 2, 40-45, doi:10.2176/nmccrj.2014-0025.
  8. Lu, K.Y.; Primus Dass, K.T.; Lin, S.Z.; Tseng, Y.H.; Liu, S.P.; Harn, H.J. N-butylidenephthalide ameliorates high-fat diet-induced obesity in mice and promotes browning through adrenergic response/AMPK activation in mouse beige adipocytes. Biochim Biophys Acta Mol Cell Biol Lipids 2021, 1866, 159033, doi:10.1016/j.bbalip.2021.159033.
  9. Yen, S.Y.; Chen, S.R.; Hsieh, J.; Li, Y.S.; Chuang, S.E.; Chuang, H.M.; Huang, M.H.; Lin, S.Z.; Harn, H.J.; Chiou, T.W. Biodegradable interstitial release polymer loading a novel small molecule targeting Axl receptor tyrosine kinase and reducing brain tumour migration and invasion. Oncogene 2016, 35, 2156-2165, doi:10.1038/onc.2015.277.
  10. Huang, M.H.; Chou, Y.W.; Li, M.H.; Shih, T.E.; Lin, S.Z.; Chuang, H.M.; Chiou, T.W.; Su, H.L.; Harn, H.J. Epigenetic targeting DNMT1 of pancreatic ductal adenocarcinoma using interstitial control release biodegrading polymer reduced tumor growth through hedgehog pathway inhibition. Pharmacol Res 2019, 139, 50-61, doi:10.1016/j.phrs.2018.10.015.

Round 3

Reviewer 2 Report

Thank you for the revision, notably adding some more explanations regarding the active agent, which significantly improved the manuscript.